# ‘Joining the Dots’: Individual, Sociocultural and Environmental Links between Alcohol Consumption, Dietary Intake and Body Weight—A Narrative Review

**DOI:** 10.3390/nu13092927

**Published:** 2021-08-24

**Authors:** Mackenzie Fong, Stephanie Scott, Viviana Albani, Ashley Adamson, Eileen Kaner

**Affiliations:** Population Health Sciences Institute, Newcastle University, Newcastle-upon-Tyne NE1 4LP1, UK; Stephanie.scott@newcastle.ac.uk (S.S.); viviana.albani@newcastle.ac.uk (V.A.); ashley.adamson@newcastle.ac.uk (A.A.); Eileen.kaner@newcastle.ac.uk (E.K.)

**Keywords:** alcohol, body weight, obesity, eating dietary intake, drinking pattern

## Abstract

Alcohol is energy-dense, elicits weak satiety responses relative to solid food, inhibits dietary fat oxidation, and may stimulate food intake. It has, therefore, been proposed as a contributor to weight gain and obesity. The aim of this narrative review was to consolidate and critically appraise the evidence on the relationship of alcohol consumption with dietary intake and body weight, within mainstream (non-treatment) populations. Publications were identified from a PubMed keyword search using the terms ‘alcohol’, ‘food’, ‘eating’, ‘weight’, ‘body mass index’, ‘obesity’, ‘food reward’, ‘inhibition’, ‘attentional bias’, ‘appetite’, ‘culture’, ‘social’. A snowball method and citation searches were used to identify additional relevant publications. Reference lists of relevant publications were also consulted. While limited by statistical heterogeneity, pooled results of experimental studies showed a relatively robust association between acute alcohol intake and greater food and total energy intake. This appears to occur via metabolic and psychological mechanisms that have not yet been fully elucidated. Evidence on the relationship between alcohol intake and weight is equivocal. Most evidence was derived from cross-sectional survey data which does not allow for a cause-effect relationship to be established. Observational research evidence was limited by heterogeneity and methodological issues, reducing the certainty of the evidence. We found very little qualitative work regarding the social, cultural, and environmental links between concurrent alcohol intake and eating behaviours. That the evidence of alcohol intake and body weight remains uncertain despite no shortage of research over the years, indicates that more innovative research methodologies and nuanced analyses are needed to capture what is clearly a complex and dynamic relationship. Also, given synergies between ‘Big Food’ and ‘Big Alcohol’ industries, effective policy solutions are likely to overlap and a unified approach to policy change may be more effective than isolated efforts. However, joint action may not occur until stronger evidence on the relationship between alcohol intake, food intake and weight is established.

## 1. Introduction

Excess body weight and heavy alcohol consumption remain two of the most intractable public health challenges globally. Worldwide, around 39% of adults have overweight and 13% have obesity [1], while one in four adults in England and Scotland regularly consume over 14 units of alcohol per week; the maximum number of units considered ‘low risk’ based on the Chief Medical Officer’s guidelines in the UK [2]. One in five drinkers in Great Britain binge drink on their heaviest drinking day (>8 units for men and >6 units for women) [2]. Obesity and alcohol misuse are leading causes of disability and disease, and disease burden attributable to elevated body mass index (BMI) and alcohol-related mortality and morbidity are greater in socioeconomically disadvantaged populations compared with more advantaged counterparts [3,4,5]. The phenomenon whereby people of lower socioeconomic position (SEP) consume the same (or less) alcohol as those from higher SEP yet experience greater alcohol-related harm is described as the alcohol-harm paradox [6]. This relationship is consistent internationally and across varying measures of SEP such as education, car ownership, income, employment, and housing tenure [7]. 

Like other nutritive beverages (mainly sugar-sweetened beverages (SSBs)), alcohol has been proposed as a causal factor of weight gain and obesity. Beverages elicit weak satiety signals relative to solid food [8,9,10] resulting in insufficient dietary compensation and greater daily energy intake [11,12]. However, compared to other nutritive beverages, alcohol has a very high energy density (29 kJ/g), providing more energy for volume than most (e.g., wine = 280 kJ/100 mL vs. cola drink = 177 kJ/100 mL). It is estimated that alcohol contributes around 10% of an adult drinker’s weekly energy intake [13] which, in the UK, roughly equates to 690–770 kJ based on data from the National Diet and Nutrition Survey showing that daily energy intake of adults 19–64 years and 65 years and over was 7690 and 6900 kJ/day, respectively [14]. This is significant given that a reduction of 400 kJ/day may attenuate population-level weight gain [15]. Further, unlike other beverages, alcohol metabolism inhibits dietary fat oxidation [16], thereby promoting body fat storage. The pharmacological properties of alcohol are posited to stimulate food intake during the drinking occasion ostensibly through integrated cognitive and hedonic mechanisms [17]. Taken together, the nutritional properties, and metabolic and psychological effects of alcohol make it a logical driver of weight gain and obesity. 

As with most research attempting to disentangle the relationship of weight with a single food/beverage item, evidence on the relationship between alcohol and weight is drawn mostly from heterogeneous cross-sectional studies, and findings are inconsistent [18]. To unpack this relationship further, researchers have investigated the role of several moderators, with sex, drinking pattern and type of alcoholic drink garnering most interest. Briefly, the hypotheses are that the relationship of alcohol and body weight is stronger: (a) in females; (b) in heavy drinkers; and (c) for beer/spirit intake. There are well-documented social and cultural differences in how men and women engage with alcohol [19,20], as well as biological sex differences in body composition, genetic factors, absorption, and metabolism [21,22]. Hypotheses regarding heavy drinking may be related to socioeconomic factors; people of lower SEP tend to binge drink more frequently than their counterparts from more advantaged backgrounds [23,24]. Similarly, drink preference may be socially patterned; wine is preferred by the middle class and is considered to be a source of cultural capital and distinction [25,26]. Wine drinking has been associated with more healthful dietary intake relative to beer drinking [27,28]. Further, standard serving sizes generally differ between drink types. In the UK, the standard serving size for beer is a 570 mL ‘pint’, providing ~880 kJ (full strength beer). For wine, the standard serving size is a 175 mL (‘medium’) glass, providing ~630 kJ. Spirits are often served with nutritive beverage mixers which contribute to greater energy content. 

Given the clear public health importance of these two interrelated issues, and the breadth of available literature, it is important to synthesise and appraise this literature to understand the knowledge base and current thinking, identify areas of further research, and inform future interventions. Thus, the aim of this narrative review was to consolidate and critically appraise the evidence on the relationship between alcohol consumption, dietary intake, and body weight within mainstream (non-treatment) populations. We acknowledge that the relationship of alcohol and weight in people with alcohol dependency (AD) is distinct. This group is more likely to experience undernutrition and have a low BMI for reasons including substitution of food for alcohol [29], changes in appetite [29,30], and interferences with nutrient digestion, absorption, and metabolism [30]. While this review focuses on general (non-treatment) populations, we discuss diverse population groups (including people with AD) in the Section 4 of this paper. As intake of both alcohol and food occurs within complex socioecological contexts, we also aimed to explore how individual, sociocultural, and environmental factors shape consumption behaviours. 

## 2. Materials and Methods

We undertook a narrative review to consolidate and critically appraise the literature exploring the impact of alcohol on food intake, the relationship of alcohol with body weight, and the individual-level mechanisms and sociocultural and environmental influences of alcohol and food intake. Publications were identified from a PubMed search using combinations of the keywords: ‘alcohol’, ‘food’, ‘eating’, ‘weight’, ‘body mass index’, ‘obesity’, ‘food reward’, ‘inhibition’, ‘attentional bias’, ‘appetite’, ‘culture’, ‘social’. We used the snowball method and citation searches and consulted articles’ reference lists to identify additional publications. Results are grouped thematically and examined below as follows: (1) The relationship between alcohol and dietary intake, and alcohol and body weight; (2) Individual-level mechanisms; (3) Social, cultural, and environmental influences on alcohol and food intake. 

## 3. Results

### 3.1. Alcohol and Dietary Intake

A recent systematic review and meta-analysis identified 22 experimental trials investigating the effect of acute alcohol intake on ad libitum food energy intake (participants are invited to eat as much or as little food as they wish) and total energy intake (energy from beverages and food) [31]. While narrative synthesis conveyed that the effect of alcohol on food energy intake was inconsistent and dependent on the comparator condition (no/negligible energy non-alcoholic drink vs. energy containing non-alcoholic drink) and the alcohol dose, meta-analysis found that food energy intake was significantly greater in the alcohol condition compared to pooled effect of energy and non-energy containing non-alcoholic beverage comparators (*n* = 12; WMD 343 kJ; 95% CI 161, 525 kJ). With regards to total energy intake, 7/8 studies found that compared to no beverage and no/negligible-energy beverage comparators, alcoholic beverages significantly increased total energy intake. Comparisons with energy-containing beverages were mixed. Meta-analysis found that total energy intake was greater in the alcohol condition compared to the energy and non-energy containing non-alcoholic/no beverage conditions (*n* = 8; WMD 1072 kJ; 95% CI 820, 1323 kJ). In keeping with these findings, analyses of nationally representative survey data in Australian adults found that energy intake for those that reported alcohol consumption on the day of the dietary recall was significantly greater than those who did not. Between-participant, age-adjusted mean daily energy intake on alcohol vs. no alcohol days was 11,409 kJ vs. 9944 kJ (*p* < 0.0001) for men, and 8303 kJ vs. 7070 kJ (*p* < 0.0001) for women, respectively. This was a mean increase of 1514 kJ (462) for men and 1227 kJ (424) for women [32]. 

There are suggestions that alcohol may impact macronutrient intake, and this may have implications on weight as macronutrients may differentially affect energy metabolism [33]. Cummings and colleagues [34] systematically reviewed the evidence on the relationship of alcohol and macronutrient intake (refined and unrefined carbohydrate/fat/protein) intake. The 18 experimental studies identified in the search yielded mixed results. While there was a trend for a single occasion of light and moderate drinking to promote fat (8/18 studies) and protein intake (5/12 studies), most studies did not detect differences in macronutrient intake in response to these drinking behaviours. Similarly, findings from 12 observational studies, most of which used nationally representative cross-sectional datasets, were also inconsistent. The most consistent findings were of the inverse association between frequent heavy drinking and refined carbohydrate intake from foods such as candies, cereal (unspecified), and chocolate (reported in 90.9% of studies), but also intake of unrefined carbohydrate from foods such as fruits, grains, and vegetables (reported in 90% of studies). Meanwhile, Parekh et al. [35] recently conducted a nuanced analysis of the Framingham Heart Offspring Cohort study (1971–2008) and explored the relationship of drinking patterns with dietary intake, and how these shifted over time as participants aged. Across all data collection points, binge drinkers consumed less fruits and vegetables and wholegrains, and had greater total fat intake than non-binge drinkers. As participants aged, total fat intake increased in binge drinkers only.

### 3.2. Alcohol and Body Weight

Given there is relatively robust evidence of the effect of alcohol on food and total energy intake compared to non-alcoholic comparator beverages, it is plausible that, without sufficient energy compensation i.e., reduction in dietary intake and/or greater physical activity, alcohol intake can cause positive energy balance. Over time, depending on the frequency and intensity of alcohol consumption, this may lead to weight gain and contribute to obesity development (Figure 1). Very few experimental studies have investigated the longer-term effect of alcohol on weight. In a cross-over trial [36], 23 healthy men with abdominal obesity drank 450 mL of alcoholic red wine and de-alcoholised wine daily for 4 weeks, randomised by sequence. There was no significant between-condition difference in body weight nor deposition of subcutaneous, abdominal, or liver fat, although a positive trend was observed in the red-wine condition for the latter (*p* = 0.09). An earlier crossover trial found that daily consumption of two 135 mL servings of red wine over six weeks did not result in differences in body weight, energy intake or macronutrient composition in free-living, healthy males compared to a 6-week abstinence phase [37]. Similarly, a crossover study in women with overweight found no significant effects of alcohol consumption (two 135 mL servings of red wine on five days/week) vs. abstinence on body weight or dietary intake over 10 weeks [38].

Sayon-Orea et al. [39] conducted the first systematic review of observational evidence on the association between alcohol consumption (all drink types) with body weight and other measures of adiposity. Of the 14 cross-sectional studies in adults, 9/14 studies (seven in men; two in women) reported a significant positive association of alcohol intake with BMI or weight gain, with a stronger positive relationship observed for heavy or binge drinking. However, a significant negative association was also observed in 9/14 studies (seven in women; two in men). For prospective cohort studies, 5/9 (three in men; two in women) showed a positive association of alcohol intake and weight gain or BMI. However, the absolute magnitude of differences was clinically important in just two studies. Associations of mixed direction were observed for cohort studies that reported measures of abdominal adiposity, although subgroup analyses in larger cohorts revealed only negative associations with wine drinking despite positive associations with total alcohol intake. One cohort study exhibited a U-shaped association between wine consumption and waist circumference, and three exhibited a J-shaped association, such that heavy (male and female) alcohol drinkers (≥28 drinks/week) showed greatest weight gain, BMI or waist circumference. Travesy and Chaput [18] conducted an updated narrative review which included additional cross-sectional and longitudinal evidence. Attention was drawn to more recent evidence that examined the relationship of alcohol intake pattern, especially heavy drinking, with adiposity outcomes. For instance, Shelton and Knott [40] analysed data from the Health Survey for England and calculated % recommended dietary intake (RDA) from alcohol on participants’ heaviest drinking day over the previous seven days. The risk of obesity was approximately 70% greater in the heaviest (≥75%RDA) vs. lightest (0–25% RDA) groups. Concordant with Sayon-Orea et al., the authors concluded that while the literature is heterogeneous and findings are inconsistent, there is a trend for the relationship between alcohol and body weight to be non-linear (J-shaped), and that drinking intensity (heavy drinking) may be a more salient behaviour than drinking frequency.

Bendsen et al. [41] focused on beer intake specifically; their systematic review included 12 experimental studies testing the effect of beer intake vs. intake of low- or non-alcoholic beer/water/no substitute beverage on anthropometric outcomes. Experimental interventions lasted between 21–126 days and, for most, participants were prescribed a certain volume of beer to drink per day or per week. Random effects pooling found that beer consumption did not increase body weight compared to the no-substitute beverage comparator (mean difference 0.54 kg; 95% CI-1.00, 2.08; *p* = 0.49; I^2^ = 0%). In contrast, consumption of alcoholic beer was found to increase body weight compared with no and low-alcohol beer (mean difference 0.73 kg; 95% CI 0.53, 0.92; *p* < 0.0001; I^2^ = 0%). Among the ten prospective cohort studies and 25 cross-sectional studies included in the review, most showed a positive relationship or no relationship between beer intake and markers of obesity; in women a negative relationship was observed in several studies. Twenty-one studies reported obesity outcomes by level of beer intake, although heterogeneity in outcome measures and beer intake e.g., frequency vs. amount, prohibited quantitative data synthesis. The cumulative raw data did not indicate a dose-response at lower or moderate beer intake (~500 mL/day), but higher beer intake (>4 L/week) may be associated positively with abdominal obesity, particularly among men. The authors noted that most studies controlled for at least some potential confounders i.e., age, education, physical activity, and smoking, but that the degree of statistical adjustment varied widely. 

Most recently, Golzarand et al. [42] conducted meta-analyses in 127 observational studies investigating the association between alcohol intake and markers of adiposity. Meta-analyses of cohort studies revealed no significant association between alcohol drinking and risk of overweight (HR 0.93; 95% CI 0.46, 1.89; I^2^ = 97.7; *p* = 0.84), obesity (HR 0.84, 95% CI 0.52, 1.37; I^2^ = 90.7; *p* = 0.48), overweight/obesity (HR 1.15; 95% CI 0.84, 1.58; I^2^ = 87.0; *p* = 0.37), nor abdominal obesity (HR 1.13; 95% CI 0.90, 1.41; I^2^ = 61.0; *p* = 0.28). In cross sectional studies, alcohol intake was associated with greater odds of having overweight (OR 1.11; 95% CI 1.05; 1.18; I^2^ = 87.7; *p* = 0.001), but not obesity (OR 1.03; 95%CI 0.95, 1.12; I^2^ = 95.1; *p* = 0.48). Subgroup analysis by alcohol dose revealed that heavy drinking (>28 g/day), but not light (<14 g/day), nor moderate drinking (14–28 g/day), was associated positively with overweight (OR 1.12; 95% CI 1.01, 1.24; *p* = 0.02). Regarding obesity, moderate drinkers had 16% lower odds of having obesity, but neither light nor heavy drinking were associated with greater odds of obesity. Those in the highest category of alcohol intake had 19% increased odds of abdominal obesity compared to those in the lowest category (OR 1.19; 95% CI 1.09, 1.29; *p* < 0.001). Heterogeneity between studies was very high and the authors rated most studies as being of very low quality/certainty based on the GRADE working group grades of evidence (i.e., the true effect is probably markedly different from the estimated effect) [43].

### 3.3. Alcohol and Weight Loss

A smaller number of studies have investigated the effect of alcohol intake on weight loss outcomes. In one randomised controlled trial (RCT), adults with overweight or obesity were prescribed hypo-energetic diets of 1500 cal/day (6300 kJ/day), of which 10% was to be consumed as either grape juice or white wine (200 mL). At 3 months follow-up, there were no significant differences in weight loss nor other anthropometric measures between groups [44]. Secondary analyses of a weight loss RCT found that while energy on no-alcohol days was significantly less than days alcohol was consumed, alcohol consumption was not associated with weekly weight loss [45]. Further, analyses of a large multicentre RCT (the Look AHEAD (Action for Health in Diabetes) study) identified that weight loss between participants who did and did not abstain from alcohol were similar during the first year regardless of study group. However, at Year 4, participants assigned to the Intensive Lifestyle Intervention (ILI) who abstained from alcohol lost 1.6% more weight relative to individuals who drank alcohol at any time during the intervention [46]. Kase and colleagues found that alcohol intake was not associated with weight before or after 26-week behavioural weight loss intervention, nor was change in alcohol intake related to change in weight [47]. However, reduction in alcohol was more relevant to weight loss for certain personality traits; specifically, those with higher levels of impulsivity experienced greater weight loss from reducing alcohol consumption.

### 3.4. Indiviudal-Level Mechanisms Linking Alcohol and Food Intake

#### 3.4.1. Appetite, Hunger and Satiety

It is proposed that alcohol stimulates appetite; a phenomenon referred to as the aperitif effect [48]. However, several studies examining the effect of alcohol on levels of ghrelin (predominant hunger-stimulating hormone), have yielded mixed findings, with some even observing a reduction in ghrelin [18,49]. It is speculated that alcohol may enhance subjective appetite, however, most studies do not indicate a significant increase in self-reported appetite following an aperitif [48]. One of the more recent studies involved participants consuming either an alcohol (0.6 g/kg) or lemonade primer, rested for twenty minutes (to allow time for alcohol absorption) and then consumed snacks ad libitum for ten minutes [50]. Compared to the lemonade group, the alcohol groups experienced greater snack urge between baseline and the rest period, and experienced a smaller decline in appetite following snack consumption.

As with other nutritive beverages, alcohol is proposed to elicit weak satiety signals relative to solid food [8,9,10,51]. Compared to food, beverages have faster gastric transit times [52,53], demand less oral processing [54,55], reduce ghrelin suppression [56,57] and elicit lower cognitive perception of anticipated satiety [56,58]. The weaker satiety responses elicited by alcohol may not stimulate sufficient feelings of fullness needed to inhibit further energy intake. 

#### 3.4.2. Inhibitory Control

Inhibitory control is an umbrella term that describes the suppression of goal-irrelevant stimuli and behavioural responses [59]. Several studies have shown that alcohol can cause deficits in inhibitory control in relation to suppression of autonomic appetitive responses [60]. A study in female undergraduate students showed that consumption of cookies following an alcohol primer (compared to an alcohol-free placebo) was mediated by inhibitory control (assessed through performance on a Stroop task), such that poorer inhibitory control was associated with greater cookie consumption [61]. Impairment of inhibitory control may be more pronounced in people who have higher trait disinhibition [50] and dietary restraint [62], although evidence is conflicted [63]. Notably, even alcohol-related cues (e.g., alcohol odour, memory elicitation of drinking) in the absence of actual consumption has been shown to reduce inhibitory control [64]. 

#### 3.4.3. Food-Related Reward and Attentional-Bias

Alcohol may stimulate greater food intake by enhancing its reward value. Studies using explicit self-report measures to assess the effect of alcohol on indices of food reward showed that intake of an alcohol primer increased: appetite, snack urge, ad libiutm intake and explicit liking of high-fat savoury foods [50,65,66]. 

It is posited that alcohol may increase attentional bias (AB) (i.e., selective attention) to food cues through classical conditioning [67] e.g., associations between an ‘aperitif’ before meals, ‘drinks and nibbles’, wine and cheese pairings etc. As AB is thought to indicate underlying appetitive motivational processes, several studies have used AB to food cues as an implicit measure of food reward. Between [68] and within-subjects [66] studies found that low dose alcohol primers (0.3–0.4 mg/kg) did not affect AB towards energy-dense foods relative to placebo. Gough et al. [66] observed that AB to food cues increased following a higher dose alcohol primer (0.6 mg/kg) compared to placebo, suggesting that these effects may only occur at higher doses. A smaller study in 23 young adults found that while low- (0.3 g/kg) and high-dose (0.65 g/kg) alcohol primers reduced alcohol-cue related AB relative to placebo, AB towards food-cues were sustained across all doses [69]. Recent evidence has indicated that alcohol (beer) odour increased attentional bias for food-cues, even in the absence of actual alcohol consumption [70].

### 3.5. Social, Cultural and Environmental Influences on Food and Alcohol Intake

#### 3.5.1. The Interconnected Role of Food and Alcohol in Social and Cultural Life

Food and alcohol ‘products’ are a source of pleasure and a valued component of social spaces (especially within the night-time economy), as well as emotional and cultural life. A qualitative synthesis of 62 papers, identified that alcohol and food were both used to overcome personal problems, facilitate fun experiences, exercise control and restraint, and demonstrate a sense of identity in 10–17-year-olds [71]. However, we have a poorer understanding of how drinking and eating behaviours interact; just a handful of studies have explored this relationship in any depth, with most data focusing on young adult populations. The first (and, to our knowledge, only) qualitative study to examine the interconnectedness of food and alcohol consumption for UK young adults, found that sociocultural, physical and emotional links between food and alcohol were an unquestioned norm for 18–25-year-olds [72]. For interviewees, eating patterns whilst drinking alcohol were tied not only to hunger, but also to sociability, traditions, and identity. Further, young adults in this study conceptualised and calculated acute risks to weight, appearance, and social status, rather than risks to their long-term health. In a study focused upon multiple health behaviours (physical activity, nutrition, weight gain), Nelson et al. [73] found that US college students suggested their weight gain resulted, in part, from drinking alcohol and alcohol-related eating, including both eating late at night after alcohol was consumed as well as eating before going out to allow themselves to consume more alcohol.

#### 3.5.2. Role of the Environment 

Physical, economic, and political environments drive the availability and affordability of food and alcohol products which, in turn, influence their consumption. In developed countries, people have virtually unfettered access to alcohol and food products high in fat, sugar and salt (HFSS). Research has demonstrated an association between alcohol outlet density (a proxy marker of physical availability) and higher alcohol consumption [74,75], and between density of hot food takeaway outlets and obesity [3,76]. Importantly, both density of fast food [77,78] and alcohol outlets [79] are associated with greater socioeconomic deprivation. The recent proliferation of digital ‘on demand’ food and alcohol delivery services has enhanced their availability and may facilitate greater consumption [80,81,82], however, this requires more investigation. Regarding affordability, both HFSS food and alcohol products are relatively inexpensive. In the UK, affordability of alcohol has increased over time and is 13% more affordable than in 2008 [83]. High strength white ciders and spirits are especially cheap and may cost as little as 19 p per unit, meaning that 14 units (the maximum number of units considered ‘low risk’ based on the UK Chief Medical Officer’s guidelines) are available for just £2.68 [84]. 

#### 3.5.3. The Role of Food and Alcohol Industries

As corporate industries, ‘Big Food’ and ‘Big Alcohol’ use similar marketing and lobbying tactics with high degrees of co-operation [85,86,87,88]. For example, both industries have engaged in the use of ‘framing’; whereby actors use discursive strategies to ‘frame’ a debate to benefit their corporate agenda [89]. Recent work by Rinaldi et al. [90] identified that different stakeholders in alcohol control (including industry) use framing discredit public health policy solutions; whilst similar studies have explored the use of framing in relation to the sugar tax [91] and food industry [92]. Petticrew et al. [93] identified the existence of a ‘cross-industry’ playbook represented by two incongruous ‘frames’: (i) aetiology of public health issues is complex, therefore, individual products cannot be blamed; and (ii) population health measures are ’too simple’ to address complex public health problems.

Both industries also use strategic advertising and marketing to promote acceptability, and even glamorisation, of convenience food and alcohol consumption. Recent research demonstrated that marketing messages by ‘Big Food’ and ‘Big Alcohol’ were both adapted in the early stages of the pandemic. Martino et al. [94] analysed the extent and nature of online marketing by leading alcohol and food/beverage brands (and their parent companies) in Australia over a 4-month period. They found that nearly 80% of posts from brands studies related to COVID-19, with quick service restaurants, food and alcohol delivery companies, alcohol brands and bottle shops most active.

## 4. Discussion

### 4.1. Appraising the Evidence

We found good evidence from pooled laboratory studies showing that acute intake of alcoholic beverages consistently and significantly increased food and total energy intake within the drinking episode. This energy appears to be additive to energy from other sources, resulting in greater daily energy intake relative to alcohol-free days. However, the cumulative findings of the four previous reviews on alcohol intake and body weight are equivocal and do not support a fully consistent relationship. That the acute effect of alcohol on food intake did not translate to a robust relationship between alcohol and weight may speak to the poor external validity of laboratory studies; responses to alcohol manifest differently in a laboratory vs. real life setting [95]. Also, Kwok et al. noted that the findings of their meta-analyses of the effect of alcohol on food intake were only generalisable to younger adults aged 18–37 years, and that significant statistical heterogeneity and small effect sizes were observed for some outcomes. 

Neither experimental nor observational evidence supported a clear association between alcohol and weight. Experimental studies were conducted in small samples and studies were likely to be underpowered to detect intervention effects. Similarly, the longest trial was conducted over ten weeks, possibly not long enough for group differences to emerge. An unclear relationship between alcohol and weight in observational evidence can be partially explained by significant heterogeneity among studies i.e., differences in assessment of drinking behaviour, cut-offs for levels of drinking, cut offs for abdominal obesity. Most epidemiological evidence was obtained from cross-sectional studies which carry the inherent limitation of not being able to determine a cause-effect relationship. Assessment of alcohol intake relies largely on self-report surveys e.g., Alcohol Use Disorders Identification Test (AUDIT-C) [96]. While these measures have adequate psychometric properties [97], they are prone to recall [98,99] and social desirability biases [100], and people tend to overestimate the amount of alcohol that constitutes a standard unit [101]. Taken together, there is a real risk for alcohol intake to be underreported which reduces confidence in studies’ findings. Recall methods e.g., the ‘Yesterday Method’—collection of detailed information about alcohol consumed the previous day—are also more likely to overestimate abstention [102]. Relating average alcohol intake at a single time point e.g., units per week, may lead to spurious findings, particularly if drinking patterns (i.e., binge) are not considered. Further, surveys may only capture current drinking behaviour, and ‘non-drinkers’ may include former drinkers who have previously experienced alcohol-related weight change; potentially reducing the validity of studies that compared drinkers to ‘non-drinkers’. 

There is also the significant potential for various factors to confound the relationship between alcohol and weight. Aside from the review by Bendsen et al. [41], others did not report the extent to which included studies statistically adjusted for confounders. Particularly pertinent is the adjustment for physical activity which could mitigate alcohol-induced energy intake. Systematic reviews have found that alcohol intake is positively (linearly or curvilinearly) associated with physical activity in young people, college-students, and the general adult population [103,104]. A cohort study in US college students found that weight motives mediated the positive relationship between heavy episodic drinking and vigorous (but not moderate) physical activity, suggesting that drinkers may engage in physical activity to compensate for additional energy intake or neutralise alcohol-related harms in line with the compensatory health behaviours model [105]. In the weight loss RCT by Carels et al. [45], duration of exercise was greater on days that alcohol was consumed, and participants who consumed alcohol more frequently had higher energy expenditure than those who drank less frequently. This may also explain why experimental studies in free-living participants found no relationship between alcohol and weight change. Another important confounding factor is ‘delayed’ dietary compensation. Studies in free-living participants have shown that corrective dietary compensation in response to deviations in daily energy intake was observed over a 24-hour period [12] and even up to three to four days later [106]. Therefore, while alcohol intake can increase daily energy intake, this additional energy may be offset in the following days. Distinctly, dietary compensation can be problematic when it manifests as ‘drunkorexia’ or ‘alcorexia’, characterised by dietary restriction, purging and/or excessive exercise to enable consumption of large quantities of alcohol and avoid weight gain and/or enhance intoxication [107,108,109,110].

Regarding the effect of moderating variables, the collective evidence gleaned from the four previous reviews of observational evidence does not support the hypotheses that drink type or sex moderates the relationship between alcohol and weight. That the effect of these moderators was inconsistent is expected given the heterogeneity and limitations mentioned previously. Notwithstanding these limitations, evidence of a J-shaped relationship between alcohol intake and weight-related outcomes (such that the positive association was strongest for heavy/binge drinking) were relatively consistent and is feasible given the substantial amount of energy consumed during a binge drinking episode. In analyses of a large national sample from England and Scotland in 18–25 year olds, Albani and colleagues [111] observed a significant positive association between alcohol consumption and BMI observed at Very High levels of intake (>75% RDA energy) in men and High to Very High intakes (>50% RDA energy) in women; equating to >1875 calories (7838 kJ) and >1000 calories (4180 kJ), respectively. This is a substantial amount of energy and would require significant dietary restriction and/or increased physical activity to mitigate the risk of weight gain. Further, drinking intensity is socially patterned and heavy/binge drinking is more prevalent in deprived groups [112,113,114]; SEP may moderate the relationship between heavy drinking and weight. This may be related to observations that those with lower SEP are more likely to engage in multiple health-risk behaviours (e.g., alcohol use, smoking, poor diet, lower physical activity) [115]. It is also suggested that heavy and/or binge drinking is linked to other behaviours implicated in weight gain. For instance, binge drinking is associated with impulsivity [116,117], which is also associated positively with energy intake and snacking, and negatively associated with diet quality [117].

### 4.2. Strengths and Limitations

This review consolidates an expansive body of literature on the relationship between alcohol intake, dietary intake, and body weight. We explored the topic from physiological, psychological, and socioecological perspectives, and included experimental, epidemiological, and qualitative evidence. Taking this comprehensive and holistic approach enabled us to ‘join the dots’ and present a nuanced and contextualised exploration of a complex issue. Given their inextricable link, we integrated evidence on food and alcohol intake to build a narrative that was more reflective of real life. In terms of limitations, neither the literature search nor article selection were conducted systematically and, therefore, selection of publications included in the review may have been biased. It is possible that some seminal papers may have been omitted unintentionally. Also, as several reviews on this topic having been conducted previously, we focused on their cumulative findings rather than those of individual studies. Therefore, there is a risk that biases, misreporting and/or misinterpretation from previous reviews may have been carried forward here. 

### 4.3. Policy and Practice Implications

Our findings have several important policy and practice implications. First, alcohol and obesity policies are typically developed and implemented independently. Given significant commonalities between these two industries (as above), a unified approach to lobbying for policy change may be more effective than siloed efforts. Also, given synergies across these two industries, effective policy solutions are likely to overlap. Knowledge exchange and collaboration between policy makers may help to optimise the effectiveness of policies and interventions. Table 1 summarises population-level food and alcohol policies/interventions across shared targets, along with selected references for further reading. We acknowledge that reducing obesity and alcohol use require a whole-systems approach and no single intervention presents a panacea. We also acknowledge that a unified approach to policy change may not occur until stronger evidence on the relationship between alcohol use, food intake and weight is available (see Section 4.4 for recommendations for future research).

Second, there are only a handful of studies examining strategies that specifically aim to reduce energy intake from alcohol. Evidence of the effect of product reformulation strategies i.e., low- and no-alcohol drinks, is limited. A recent study found that low-alcohol products are perceived to target non-traditional consumers (pregnant women) and occasions (weekday lunchtimes), suggesting potential challenges to their uptake [159]. In a recent systematic review and meta-analysis, Robinson et al. [153] found moderate evidence that consumers are unaware of the energy content of alcoholic drinks, and that consumers support energy labelling. However, most studies found no effect of energy-labelling on actual or intended alcohol consumption; studies were generally of poor methodological quality and none were conducted in a real-life setting. Displaying the energy content of alcohol may result in complex public health messaging and disordered ‘drunkorexia’ or ‘alcorexia’ behaviours e.g., binge drinking, restriction of dietary intake/purging. These behaviours increase the likelihood of intoxication, result in blood alcohol levels rising sharply affecting the brain and subsequent behaviour, which in turn steeply increases the risk of acute harm such as from accidents. Some [160,161,162] but not all [163] studies suggest that such weight control behaviours are particularly prevalent amongst females and can lead to wider repercussions for health and wellbeing. For example, amongst UK women, consumption of alcohol without food is associated with higher risk of liver cirrhosis [164]. Meanwhile, in a prospective cohort study of UK Biobank (UKB) participants, Jani et al. [165] found a 10% higher risk of mortality with alcohol drinking without food compared to alcohol drinking with food amongst 38–73 year-olds. 

Third, policy and practice strategies are more challenging to implement in marginalised or at-risk populations, such as those in alcohol recovery or experiencing homelessness or food-insecurity. Here, the evidence-base becomes sparser, with a small pool of quantitative research and a dearth of qualitative data. Again, most studies do not focus explicitly on this relationship in marginalised/at risk groups; usually this is discussed in the context of wider aims/objectives or with one product given a peripheral mention in the context of the other. Thus, Puddephatt et al. [166] found that a small minority of UK food bank clients reported using alcohol and other illicit substances to suppress hunger and to cope with the stress of their access to food. Meanwhile, Reitzel et al. [167] identified that, among US homeless adults, 28.4% of the sample had probable alcohol dependence, 25% were heavy drinkers, and 78.4% were food insecure. Further, heavy drinking and probable alcohol dependence/abuse were each associated with increased odds of food insecurity. Similar associations were identified by Bergmans et al. [168]. Tan and Johns [169] did specifically focus on the links between alcohol use and eating disorders amongst those alcohol dependent or in early recovery, and found the ‘alcohol wheel’ to be triggering for those more susceptible to an eating disorder in alcohol services. Further, Thomson and Paudel (2018) highlight the possibility of addiction swap (substitution of sugar for alcohol addiction) for those in alcohol recovery. 

### 4.4. Future Research

Despite no shortage of observational research, the relationship between alcohol and weight remains uncertain. To genuinely improve our understanding of this relationship, rather than gratuitously generating further cross-sectional analyses, resources should be prioritised for research that uses innovative methodologies to make a novel and meaningful contribution to the literature, and that considers the complexities of the relationship and important variables linking food and alcohol consumption (regular drinking pattern and previous drinking status, physical activity levels, personality traits, health motives, SEP). Rather than examining the relationship of alcohol intake and weight which may be an oversimplification even with statistical adjustment, a latent class analysis may be more illuminating, given that weight-related behaviours and food/beverage consumption patterns tend to cluster. Methodologies that sample behaviour in the field such as ecological momentary assessment (EMA) [170] or diary studies [106] that are conducted over several weeks would provide valuable, richly detailed data on alcohol use and energy balance behaviours in real life. Similarly, more qualitative work is needed to improve our understanding around consumers’ knowledge, perceptions and attitudes regarding alcohol, food, weight, and weight-related behaviours across the life course. This would help to identify and develop targeted interventions that are informed by real life experiences. Also, youth alcohol consumption has declined steadily and significantly over the last 15 years in many high-income countries including the UK, with those young people who do drink starting to do so at a later age, less often and in smaller quantities [171,172]. What, then, are the implications of this on eating behaviours and body weight? How consumers use no- and low-alcohol products requires further research i.e., who uses them, how are they used and for what reasons, are they substitutive or additive to the whole diet, what are facilitators and barriers to their uptake, do they lead to self-licensing (e.g., I had a low-alcohol beer, therefore, I can eat more chips). More generally, all future research must adequately represent people from marginalised and disadvantaged communities who are disproportionally affected by alcohol misuse and obesity.

## 5. Conclusions

Alcohol appears to increase food and energy intake within a drinking episode, and this energy increases daily energy intake. However, as with other research that attempts to disentangle the relationship of weight with a single nutrient/food/beverage/meal, evidence is equivocal and does not support a fully consistent relationship. Most evidence is derived from cross-sectional survey data and is limited by methodological weaknesses and heterogeneity. That the evidence remains uncertain despite no shortage of observational research indicates that more nuanced analyses and creative research methodologies are needed to capture what is clearly a complex and dynamic relationship. To meaningfully advance our understanding of the relationship between alcohol, food and weight, future research must account for relevant behaviours and variables and, importantly, how they cluster with patterns of alcohol intake e.g., binge drinking, and how they shift across the life course. Qualitative work and sampling weight-related behaviour and alcohol use in the field (e.g., using EMA protocols or diary studies) would add much needed richness and nuance to the literature. Given synergies between ‘Big Food’ and ‘Big Alcohol’ industries, effective policy solutions are likely to overlap and a unified approach to policy change may be more effective than isolated efforts. However, joint action may not occur until stronger evidence on the relationship between alcohol intake, food intake and weight is established. 

## Figures and Tables

**Figure 1 nutrients-13-02927-f001:**
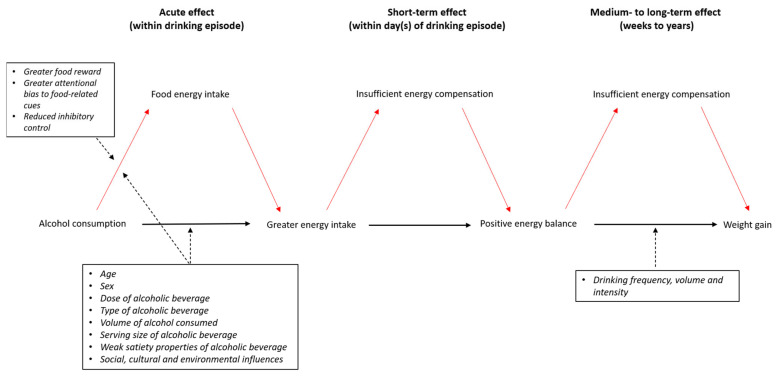
The hypothesised causal pathway from alcohol intake to weight gain. The red lines represent a proposed mediational relationship, and the dotted black lines represent proposed moderation. Body weight and energy compensation are influenced by a significant number of factors; these are beyond the scope of this review and have intentionally been omitted from this figure.

**Table 1 nutrients-13-02927-t001:** Population-level food and alcohol policies across shared targets/points of intervention.

Target of Policy/Strategy	Foods and Non-Alcoholic Beverages	Alcoholic Beverages
Fiscal policy	Taxation of sugar content in SSBs [118,119], and dietary fat content of foods [120]	Minimum unit pricing [121,122,123] Taxation of alcohol content [124,125]
Mass media and marketing	Restrictions on advertising [126] e.g., UK government’s 9 pm watershed on television advertising of food and drink products high in fat, sugar and salt (HFSS) [127] Regulation of marketing e.g., HFSS food companies’ sponsorship of, and advertising in sport environments [128,129]	Restrictions on advertising [130,131] Regulation of marketing e.g., alcohol sponsorship of, and advertising in sport environments [132,133]
Sales availability	Regulation of takeaway outlet density e.g., exclusion zones around schools [134], planning regulations [135,136] Regulation of digital on demand food delivery services [137]	Regulation of alcohol outlet density [138,139] Regulation of licensing hours [138,140] Regulation of digital on demand alcohol delivery services [141]
Product server setting	Information-based cue at point-of-purchase e.g., grocery store [142,143] Regulation of volume-based price promotions e.g., ‘2 for 1 deals’ [144]	Information-based cue at point-of-purchase e.g., bar [145] Regulation of price and volume promotions e.g., ‘2 for 1 deals’ [146,147]
Product reformulation	Production of reduced fat and sugar product varieties [148,149]	Production of low- and no-alcohol beverage alternatives [123,150]
Product labelling	Use of on-pack nutrition labelling [142] Use of on-pack health warning labels [151,152]	Use of on-bottle energy content labelling [153] Use of on-bottle health warning labels [152,154]
Standard serving sizes	Reduction of standard serving sizes [155,156]	Reduction of standard serving sizes [157,158]

## Data Availability

Not applicable.

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
