# Peer review of "‘Joining the Dots’: Individual, Sociocultural and Environmental Links between Alcohol Consumption, Dietary Intake and Body Weight—A Narrative Review"

_nutrients, 2021, doi:10.3390/nu13092927_

Round 1
Reviewer 1 Report
Here is a narrative review on the main problem of alcohol consumption.
The authors analyzed the literature profoundly and try to investigate a possible correlation between alcohol consumption, calorie intake, and weight gain; moreover, they explored the association of alcohol intake with individual, sociocultural, environmental factors.
The authors’ principal findings are that: evidence is equivocal and does not support an entirely consistent relationship between alcohol intake and food and energy intake. Most evidence is derived from cross-sectional survey data and is limited by methodological weaknesses and heterogeneity. Food and alcohol have an interconnected role in social and cultural life. Environment and alcohol industries play an essential role. Effective policy solutions are likely to overlap, given synergies between ‘Big Food’ and ‘Big Alcohol’ industries. Knowledge exchange, collaboration, and unification between policymakers lobbying for action within these two industries may be more effective than solo efforts.
The principal limitation of the study is that neither the literature search nor article selection was conducted systematically, and, therefore, the choice of publications included may have been biased.
A readable table enriches the work.
The work was a bit long to read due to the vastness of the aspects covered. However, it is very well articulated and divided into subheadings to create a reading order.
This work is of estimable value because alcohol consumption and obesity are significant problems for society.
Minor issues:
- remove lines 110-112
- chek 9/14 on lines 174 and 177 (the sum does not seem fair)
Author Response
Response to Reviewer 1 Comments
Thank you for time and expertise to review our manuscript. We appreciate your constructive feedback and are grateful for your positive comments and support. Please see below responses to your comments.
Point 1:
Remove lines 110-112
Response 1:
Thank you for picking up this oversight. These lines from the template have now been deleted.
Point 2:
chek 9/14 on lines 174 and 177 (the sum does not seem fair)
Response 2:
We agree the results appear confusing/incorrect. However, this was checked against the original article and is in fact correct, it just depends on the sex of the sample. We have included the sex of the samples to provide more clarity:
“Of the 14 cross-sectional studies in adults, 9/14 studies (seven in men; two in women) reported a significant positive association of alcohol intake with BMI or weight gain, with a stronger positive relationship observed for heavy or binge drinking. However, a significant negative association was also observed in 9/14 studies (seven in women; two in men).”
Reviewer 2 Report
General comments
This is a very well-constructed and well-written narrative review, which captures key elements in the relationship between alcohol intake, energy and weight increases and highlights important flaws in the research as to why it might not capture this relationship seen in real life settings. I congratulate the authors on the high standard of this work. I only have very minor comments.
Specific (minor) comments
Line 259, spelling error: change ‘mechansisms’ to ‘mechanisms’
Line 341, separate ‘overtime’ into two words, ‘over time’.
Line 373, grammar. Please revise sentence ‘to poor the external validity of laboratory studies;’
Line 483. Please change ‘in real-life setting’ to either ‘in a real-life setting’ or ‘in real-life settings’.
Lines 463-467. This comment is a valid one ‘…a unified approach to lobbying for policy change may be more effective than siloed efforts’. However, because of the existing body of evidence not substantiating the relationship between alcohol intake and weight gain, this unified approach is unlikely to happen. Perhaps it would be valuable to mention this here, emphasising that with better and more innovative research to verify these relationships, this unified approach in policy change might have a chance of occurring. This is not a ‘must change’, but rather a suggestion if the authors feel it enhances this paragraph.
Author Response
Response to Reviewer 2 Comments
Thank you for time and expertise to review our manuscript. We appreciate your constructive feedback, and are grateful for your positive comments and support. Please see below responses to your comments.
Point 1: Line 259, spelling error: change ‘mechansisms’ to ‘mechanisms’
Response 1: This has been changed to ‘mechanisms’.
Point 2: Line 341, separate ‘overtime’ into two words, ‘over time’.
Response 2: This has been changed to ‘over time’.
Point 3:
Line 373, grammar. Please revise sentence ‘to poor the external validity of laboratory studies;’
Response 3: This has been changed to ‘to the poor external validity of laboratory studies’.
Point 4:
Line 483. Please change ‘in real-life setting’ to either ‘in a real-life setting’ or ‘in real-life settings’.
Response 4: This has been changed to ‘in a real-life setting’.
Point 5:
Lines 463-467. This comment is a valid one ‘…a unified approach to lobbying for policy change may be more effective than siloed efforts’. However, because of the existing body of evidence not substantiating the relationship between alcohol intake and weight gain, this unified approach is unlikely to happen. Perhaps it would be valuable to mention this here, emphasising that with better and more innovative research to verify these relationships, this unified approach in policy change might have a chance of occurring. This is not a ‘must change’, but rather a suggestion if the authors feel it enhances this paragraph.
Response 5: Thank you for this feedback. We agree that without substantiating evidence, a unified approach to policy change is unlikely to occur. We have acknowledged this at the bottom of this paragraph (Lines 470-474): “We acknowledge that reducing obesity and alcohol use require a whole-systems approach and no single intervention presents a panacea. We also acknowledge that while the evidence on the relationship between alcohol use and weight remains uncertain, a unified approach to policy change may not occur until stronger evidence is available (see 4.4 Future research).”
This is also reflected in the conclusion (Lines 562-566) and abstract (Lines 28-31): Given synergies between ‘Big Food’ and ‘Big Alcohol’ industries, effective policy solutions are likely to overlap and a unified approach to policy change may be more effective than isolated efforts. However, joint action may not occur until stronger evidence on the relationship between alcohol intake, food intake and weight is established.